# Study on Mechanical Properties of Deep Expansive Soil and Coupling Damage Model of Freeze–Thaw Action and Loading

**Zhuliang Zhu, Bin Lin * and Shiwei Chen**

School of Civil Engineering and Architecture, Anhui University of Science and Technology, Huainan 232001, China; zhuzl1999@163.com (Z.Z.)

* Correspondence: linbin8910@163.com

**Abstract:** This study is primarily intended to present a damage constitutive equation under the combined action of confining pressures and freeze–thaw cycles by subjecting deep expansive clay to the consolidated undrained triaxial tests. We study the influence of the numbers of freeze–thaw cycles on various mechanical indexes of soil by using the TSZ-2 instrument (fully automatic triaxial instrument). As the number of freeze–thaw cycles increases, the ultimate peak stress of the soil decreases, and then, the effect of the freeze–thaw effect on the shear strength gradually weakened. By combining the expression method of the damage variable under the action of loading alone with the expression method under the action of freeze–thaw cycles alone, we brought in the damage evolution equation to obtain the damage constitutive equation under the combined action of confining pressures and freeze–thaw cycles. The stress values under three confining pressures (100 kPa, 200 kPa, and 300 kPa) can be determined by using the final damage constitutive model. The measured data with a water content of 17% and six freeze–thaw cycles were compared with the theoretical data. The actual strength values were 118.4 kPa, 152.3 kPa, and 184.1 kPa, and the theoretical strength values were 120 kPa, 150 kPa, and 186 kPa. The fitting degree of the strength value was as high as 99%, which verifies the feasibility of this model. This study can serve as an available reference for well wall construction and disaster prediction in deep coal mining.

**Keywords:** frozen soil; deep clay; triaxial tests; freeze–thaw cycles; damage model

## 1. Introduction

Coal resources make up a large proportion of the world's energy supply. Currently, coal accounts for 27% of the primary energy consumption worldwide [1]. Some countries around the world have an increasing demand for energy resources [2,3], while coal resources cannot be substituted by alternative sources of energy. The extraction of shallow coal reserves has been insufficient to satisfy the rising demand, and consequently, there has been a gradual increase in the excavation depths of mines worldwide [4,5]. With the increase in the depth of mining, the geological conditions are becoming more complex, landslide disasters caused by various reasons are prone to occur [6–9], and the mining environment is becoming increasingly unfavorable [10]. So far, artificial ground freezing has become the way to prevent water inrush and well gushing in the process of shaft sinking in deep alluvium and fractured rock strata [11–13]. The artificially chilled brine passes through the freezing pipe, and consequently, the frozen wall is formed around the shaft. The frozen wall is a useful protective screen for resisting earth pressure and preventing groundwater infiltration into the well. The construction complexity of the frozen wall inevitably increases with soil thickness and depth of freezing, resulting in more frequent disasters caused by the freezing and melting process of soil [14,15]. Therefore, it is imperative to study the mechanical properties and damage progression of deeply frozen soil.

From the perspective of damage mechanics, the failure of soil resulting from the application of external loading is a transition from primary to secondary soil structure.

The evolution of damage can be seen as a blend of undisturbed soil and soil that is wholly damaged [16–18]. Kachanov et al. [19] first proposed the concept of damage mechanics in the study of creep fracture and cited the concept of the 'continuity factor'. Shen [20] pointed out that, when the external loading is applied to the soil, the soil can be regarded as the superposition of the undisturbed soil and the damaged soil, and he introduced the concept of a damage ratio and established the damage constitutive model of structural clay. Tong et al. [21] used cement to improve the soil, studied the damage evolution of the improved soil, and established the damage constitutive model of cement-improved soil under uniaxial compression, based on the study of the evolution law. Zhu et al. [22] used Weibull distribution to simulate the whole process of damage evolution, coupled stress field, temperature field, and water field and fitted the stress–strain curves of frozen sand at different temperatures to obtain the model. Zhang et al. [23] conducted the rapid loading test, consolidation test, and triaxial compressive strength test of the natural sedimentary structural clay in Lianyungang and studied the relationship between the damage variable and disturbance degree of soil. Shi et al. [24] proposed the damage variable considering the cutting effect of the ice lenses and established the corresponding elastoplastic damage constitutive model of the frozen soil. Sun et al. [25] proposed an elastoplastic damage constitutive model based on super/sub-loading yield surface and verified the model by observing the mechanical behavior of frozen soil and frozen saline soil through the triaxial compression test. He et al. [26] established a new composite structure fractional constitutive model by establishing the creep relationship between the bonding unit and the friction unit based on the microscopic damage mechanism of frozen soil. The above research shows that scholars have limited research on the damage evolution of soil to only from a single direction, and there is almost no research on deep soil.

Therefore, in this paper, deep expansive clay was studied, and the damage evolution of soil under the combined action of loading and freeze–thaw cycles was studied in more depth. The conventional triaxial test and triaxial shear tests under the freeze–thaw cycles of deep expansive clay were carried out, and the influence of the number of freeze–thaw cycles on various mechanical indexes of soil was studied. The expression method of the damage variable under the action of loading alone and the expression method under the action of freeze–thaw cycles alone were coupled and brought into the damage evolution equation to obtain the damage constitutive equation under the combined action of confining pressures and freeze–thaw cycles and compared with the actual stress–strain curve. This study can provide some reference value for well wall construction and disaster prediction in deep coal mining.

## 2. Experiments Design

### 2.1. Sample Preparation

The deep expensive clay samples were obtained from detection hole of an air shaft in a mining area of Anhui, east China. The basic physical properties of the soil are listed in Table 1, and the soil particle size distribution was calculated according to ASTMD422-63 [27] and listed in Table 2.

**Table 1.** Basic physical property parameters of soil.

| Natural Moisture Content (%) | Natural Density (g·cm$^{-3}$) | Dry Density (g·cm$^{-3}$) | Specific Gravity | Liquid Limit (%) | Plastic Limit (%) | Free Expansion Rate (%) |
|---|---|---|---|---|---|---|
| 19.31 | 1.965 | 1.647 | 2.467 | 39.33 | 18.77 | 58.54% |

**Table 2.** Particle size grading table.

| Particle Size (mm) | Particle Size Ratio (%) |
|---|---|
| <0.075 | 12.23 |
| 0.075~0.25 | 37.97 |
| 0.25~0.5 | 30.19 |
| 0.5~1.0 | 13.64 |
| 1.0~2.0 | 5.7 |

The soil sample preparation method was as follows: the bulk disturbed soil was crushed by the soil crusher and sieved through a 2 mm standard sieve. The soil samples were dried in an oven at 105 °C for 8 h and cooled to room temperature in a sealed container. According to the natural water content of the soil, the experimental water content was set reasonably. A certain amount of water was added evenly to the soil samples by a sprayer to achieve the moisture content targets of 14%, 17%, 21%, and 24%. In order to ensure even distribution of water in the soil, soil samples were then sealed in a closed container for 24 h. After that, some soil samples were taken for moisture content testing to ensure that the target moisture content was achieved. Five equal amounts of soil were added to the standard mold, the sample being compacted after each addition. In order to reduce the friction between the soil samples and the mold, Vaseline was used to coat around the mold. Finally, a number of soil samples with an inner diameter of 39.1 mm and a height of 80 mm were produced. In addition, the soil samples were immediately sealed with plastic film [28,29].

### 2.2. Experimental System

As shown in Figure 1, the test instrument was the TSZ-2 fully automatic triaxial instrument produced by the Nanjing Soil Instrument Factory. The axial load of the upper and lower limit device adopts the force sensor; the measurement range is 0–30 kN, and the accuracy is ±1%. The freeze–thaw cycle system is a high–low temperature and humidity constant test chamber produced by Taiwan, China. It has the function of PID automatic calculus and high precision.

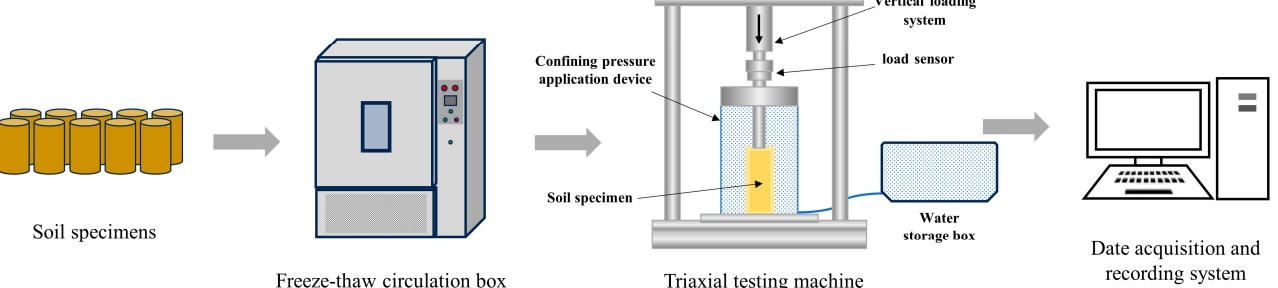

**Figure 1.** Schematic diagram of triaxial shear test under freeze–thaw cycles.

### 2.3. Experimental Program

The triaxial consolidation undrained method was used in the test. The axial loading rate was set to 0.8 mm/min during the test, and the confining pressure was controlled at 100 kPa, 200 kPa, and 300 kPa, respectively. If the stress–strain curve had no peak value, the stress corresponding to 15% strain was the ultimate stress. After completing the conventional triaxial test, we placed the prepared soil samples in the high–low temperature and humidity constant test chamber produced for freeze–thawing and the time of freezing and thawing was controlled at 4 h to ensure that 3 freeze–thaw cycles was 1 test cycle. Five test cycles of freeze–thaw cycles, i.e., 0, 3, 6, 9, and 12, were considered. After the

freeze–thaw cycle, the triaxial shear test under freeze–thaw cycles was carried out. The test method and confining pressure setting were the same as the conventional triaxial test.

## 3. Experiments Results and Analysis

The stress–strain curve obtained from the consolidation undrained test is shown in Figure 2. From the stress–strain curve of deep clay under different water contents, it can be seen that the lower the water content, the greater the compressive strength of the soil. The compression curve of deep clay under four water contents undergoes a linear elastic stage and a nonlinear stage. In the case of low water content, the samples under triaxial confining pressure were all strain softening. The higher the water content, the later the peak point appeared. The soil sample with low water content had a peak point before the strain was 20%, indicating that the curve of deep clay showed a strain-hardening type when the water content gradually increased. At the same time, it can be seen from the figure that the stress–strain curve of deep expansive clay under high confining pressure tended to strain hardening, while low confining pressure showed strain softening.

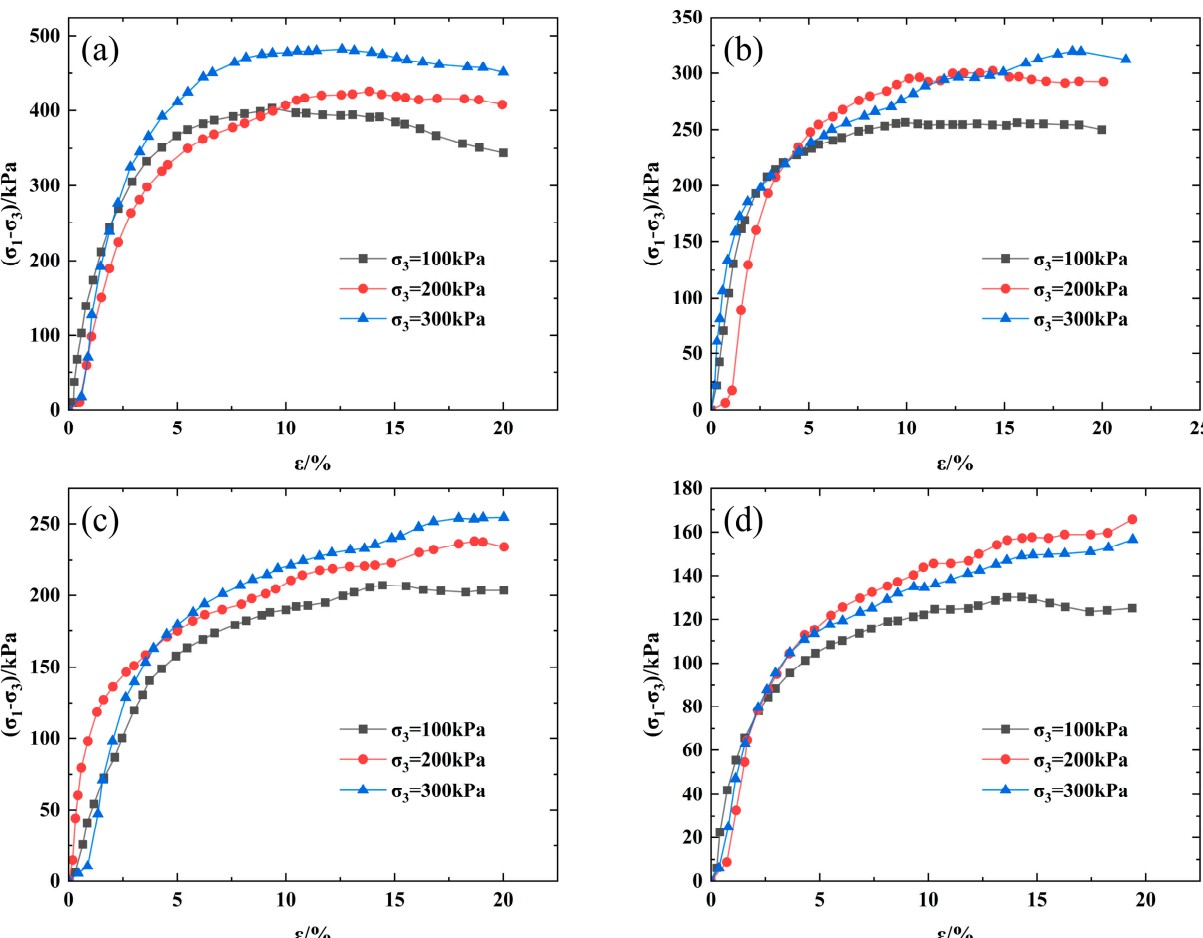

**Figure 2.** The stress–strain curve under different moisture contents: (**a**) ω =14%, (**b**) ω = 17%, (**c**) ω = 21%, and (**d**) ω = 24%.

Figure 3 illustrates that under the same confining pressure, the growth rate of stress with strain gradually decreased as the freeze–thaw times increased. Meanwhile, the ultimate peak stress also decreased as a result of the freeze–thaw cycles. The initial stress was linearly proportional to the strain during triaxial compression and, then, gradually became a nonlinear relationship because of internal damage. Under different freeze–thaw cycles, the stress–strain curves of triaxial tests all showed strain hardening. Under the three confining pressures, the relationship between compressive strength and number of freeze–thaw

cycles is shown in the Figure 4. It can be seen that the change rules were basically similar under the three confining pressures. As the number of freeze–thaw cycles increased, the compressive strength decreased significantly. In addition, the compressive strength value decreased significantly when the number of freeze–thaw cycles was small. When the freeze–thaw cycles reached a certain number, the compressive strength remained basically stable, and the change was small. It can be concluded that as the number of freeze–thaw cycles increased, the effect of the freeze–thaw effect on the compressive strength gradually weakened.

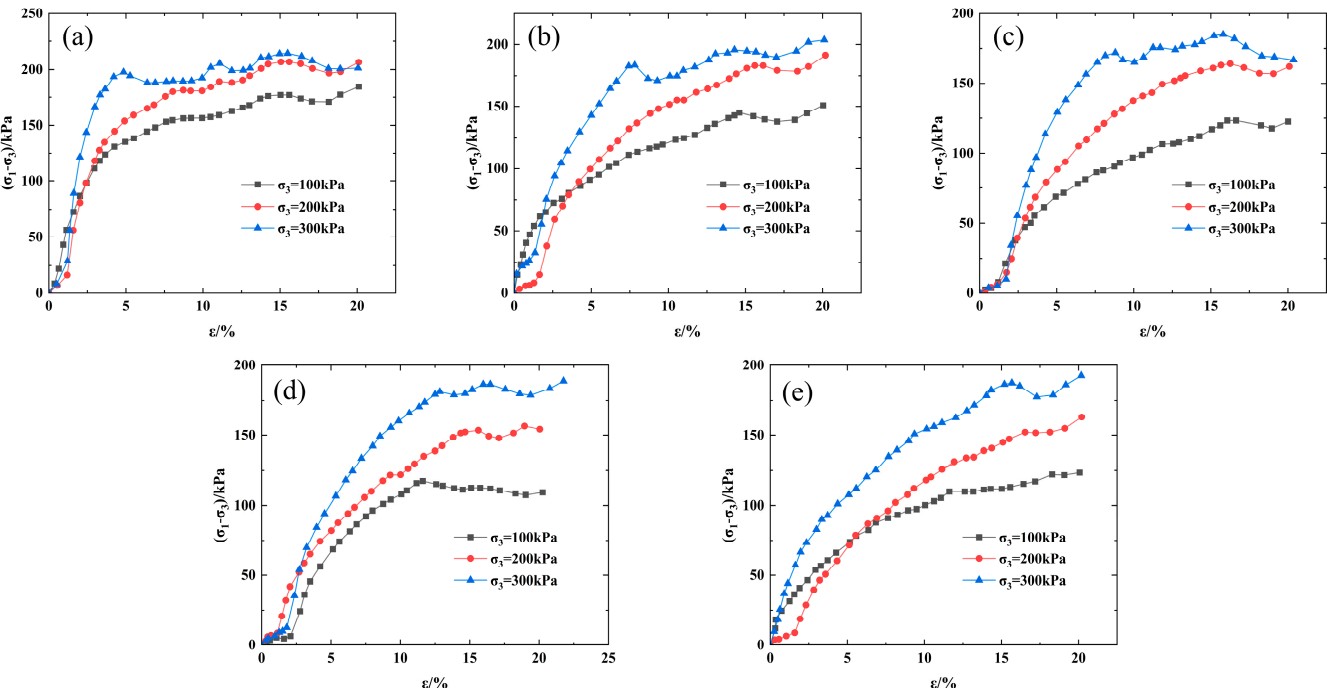

**Figure 3.** Stress–strain curves under different numbers of freeze–thaw cycles: (**a**) 0 time, (**b**) 3 times, (**c**) 6 times, (**d**) 9 times, and (**e**) 12 times.

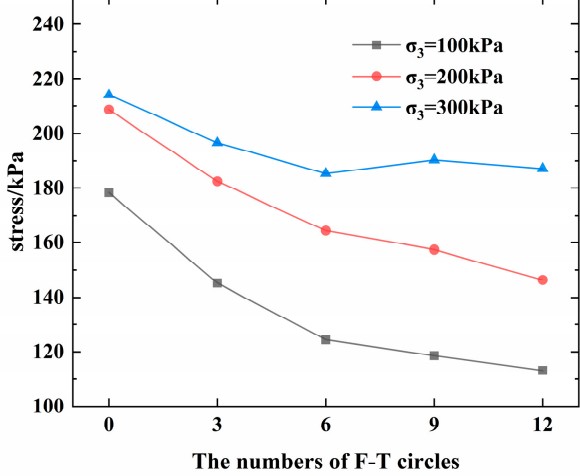

**Figure 4.** The relationship between compressive strength and number of freeze–thaw cycles.

The value of cohesion c and internal friction angle φ are the basic physical index of the triaxial test. The Mohr stress circle envelope was drawn by the limit stress and strain of each freeze–thaw cycle, and the value of the internal friction angle and cohesion was calculated. The Mohr stress circle envelope in several states is shown in Figure 5.

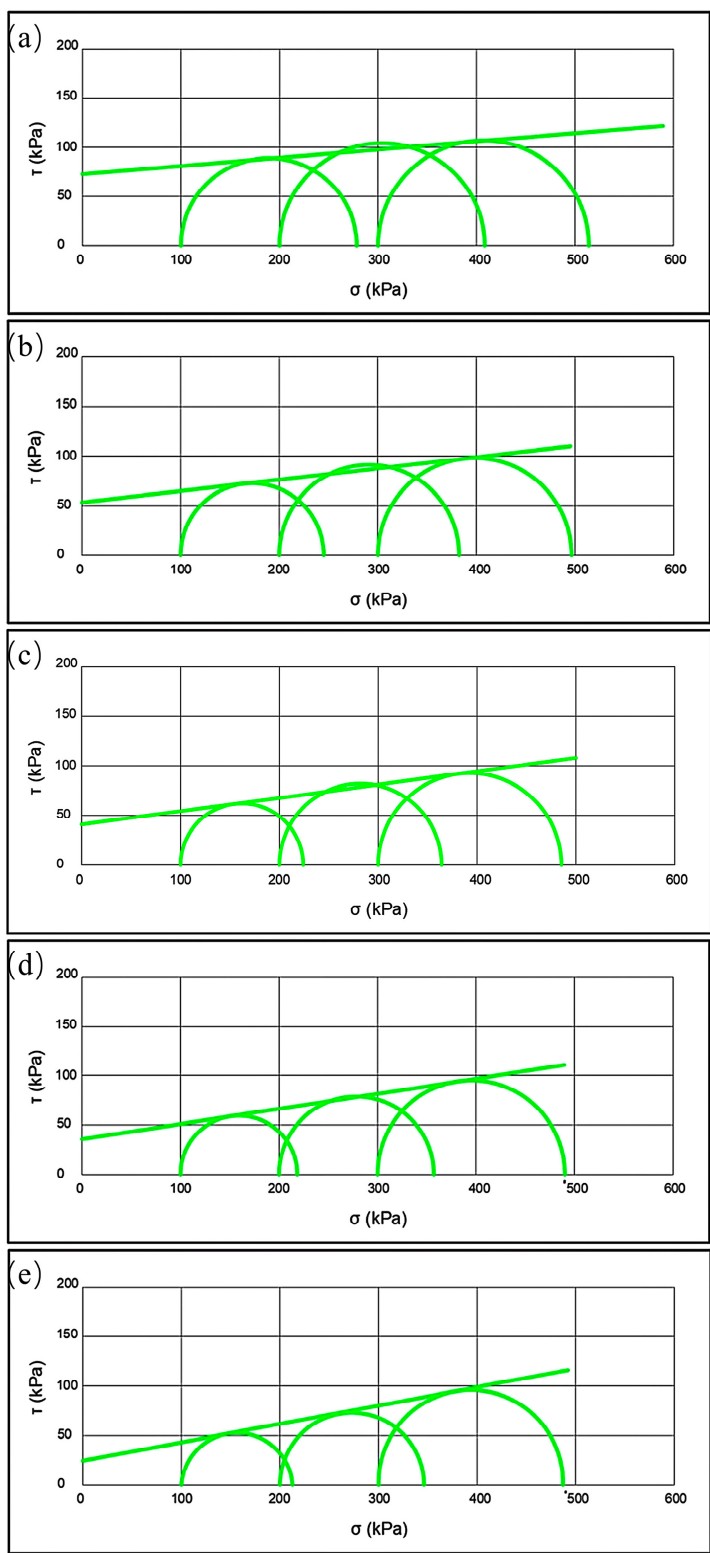

**Figure 5.** The Mohr stress circle under different numbers of freeze–thaw cycles: (**a**) 0 time, (**b**) 3 times, (**c**) 6 times, (**d**) 9 times, and (**e**) 12 times.

It can be seen from Figure 6 that as the number of freeze–thaw cycles increased, the cohesion tended to decrease, while the internal friction angle tended to increase. For small numbers of freeze–thaw cycles (within six cycles), the freeze–thaw cycles greatly affected both cohesion and the internal friction angle. When the number of freeze–thaw cycles

reached roughly nine times, cohesion and the internal friction angle generally stabilized, indicating little impact from freeze–thaw cycles on these shear-strength indices.

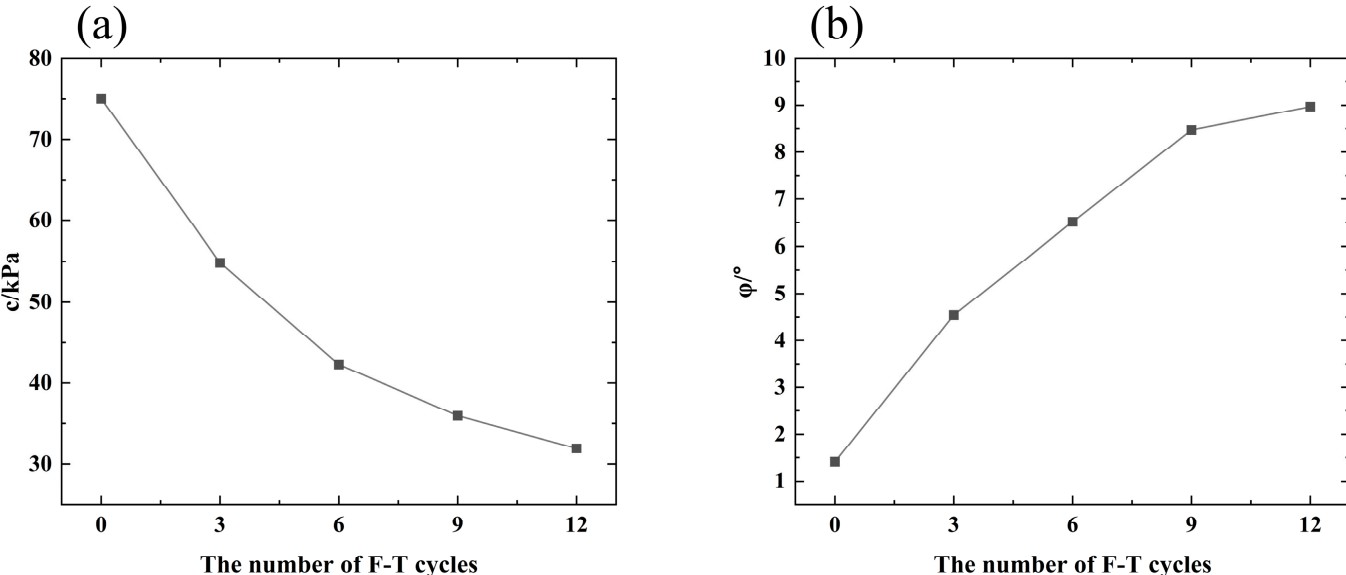

**Figure 6.** The change diagram of mechanical index with the number of freeze–thaw cycles. (**a**) The cohesion, (**b**) The internal friction angle.

## 4. Damage Constitutive Model of Deep Expansive Clay under Freeze–Thaw Cycles

The soil subjected to freeze–thaw cycles is damaged under the combined action of its own structural defects and internal microcracks and external loads during the compression process, which will lead to the propagation and interconnection of damage so that the macroscopic mechanical properties of freeze–thaw soil are significantly weakened. In addition, due to the repeated freeze–thaw action, small cracks in the soil can develop, which has a huge impact on the structure of the soil. How to properly reflect the damage law of soil under the combined action of loading and freeze–thaw cycles, the key is to select the appropriate damage variable. Many scholars had selected micro-benchmarks for damage analysis under loading [30,31]; however, there were some defects. The reason is that the initial soil damage has a great degree of randomness and random evolution. If the constitutive relationship of traditional methods is used, the randomness of soil in the damage process cannot be well reflected. Some scholars had cited the idea of probability and statistics [32], which provided a new way to solve such problems.

### 4.1. Establishment of Damage Variables

The effective stress is the main factor affecting the strain of the soil body. When the material damage occurs, its constitutive relationship only needs to change the stress to the effective stress, which is the famous Lenaitre's strain equivalence hypothesis, so as to establish the basic equation of the damage constitutive model of clay expansive soil:

$$\sigma_{ij} = \sigma^*(1 - D) = E_{ijkl}\varepsilon_{kl}(1 - D) \tag{1}$$

where $E_{ijkl}$ is the tensor component of material elastic stiffness, $\sigma_{ij}$ is the stress tensor component, $\sigma^*$ is the effective stress tensor component, $\varepsilon_{ij}$ is the strain tensor component, and $D$ is the damage variable.

In soil mechanics, soil is often considered a continuous medium from a macroscopic perspective. Due to the dispersion of soil particles, soil is also considered to be a porous medium. However, there are many kinds of soil particles and cement in it, and the existence of interlocking between soil particles makes the soil have good integrity and structure.

Therefore, it is considered that the soil is continuous during a loading process of little structural damage, so the following assumptions are put forward:

1.  When the sample is subjected to the loading, macroscopically, it can be regarded as an isotropic microelement, while at the microscopic level it contains some of the basic information that causes damage, and at this point, it can be regarded as a nonhomogeneous microscopic material.

2.  The clay microelement can be regarded as linearly elastic before damage is produced under loading. At this time, its strength follows the Hooke law, and the nonlinearity of stress–strain results from the generation of material damage. The initial tangential modulus can also be replaced by the elastic modulus of the undamaged material.

3.  The damage caused by microelements is considered to be strength damage in clay materials.

At the microscopic perspective, the ratio of the area in a damaged unit to the total cross-sectional area is called the damage variable. Macroscopically, it can be defined as the ratio of the number of self-damaged units, $N_f$, to the total number of cells, $N$. The damage equation is expressed as follows:

$$D = \frac{N_f}{N} \tag{2}$$

The former scholars demonstrated that the Weibull distribution is more appropriate to describe the pattern of strength distribution in clay under freeze–thaw cycles [33,34]. By introducing elemental intensity as a random variable and combining it with the Weibull distribution function, the following can be obtained:

$$\begin{cases} P(F^*) = \frac{\alpha}{F_0} \left( \frac{F^*}{F_0} \right)^{m-1} \exp\left[ -\left( \frac{F^*}{F_0} \right)^m \right], & F^* \geq 0 \\ 0 & F^* \leq 0 \end{cases} \tag{3}$$

The damage variable $D$ can be expressed as follows:

$$D = \frac{N_f}{N} = \int_{-\infty}^{F} P(x)dx = 1 - \exp\left[ -\left( \frac{F^*}{F_0} \right)^m \right] \tag{4}$$

where $m$ and $F_0$ are parameters of the Weibull distribution. They are determined by the mechanical properties of the clay.

### 4.2. Determination of the Clay Microelement Strength

The Druck–Prager damage criterion can be used to calculate the microelement strength of clay, and the basic form of clay microelement damage is selected as follows:

$$F^* = \alpha I_1^* + \sqrt{J_2^*} \tag{5}$$

In this formula,

$$I_1^* = \sigma_1^* + \sigma_2^* + \sigma_3^* \tag{6}$$

$$J_2^* = \frac{1}{6}\left[ (\sigma_1^* - \sigma_2^*)^2 + (\sigma_2^* - \sigma_3^*)^2 + (\sigma_1^* - \sigma_3^*)^2 \right] \tag{7}$$

The damage evolution equation can be obtained by substituting Equation (5) into Equation (4), resulting in the following:

$$D = 1 - \exp\left[ -\left( \frac{\alpha I_1^* + \sqrt{J_2^*}}{F_0} \right)^m \right] \tag{8}$$

Equation (9) is obtained based on the previous assumptions about the damage variables and then combined with the Lemaitre strain equivalence principle as:

$$\frac{1}{1-D} = \exp\left[\left(\frac{\alpha I_1 + \sqrt{J_2}}{F_0(1-D)}\right)^m\right] \tag{9}$$

where

$$I_1 = \sigma_1 + \sigma_2 + \sigma_3 \tag{10}$$

$$J_2 = \frac{1}{6}\left[(\sigma_1 - \sigma_2)^2 + (\sigma_2 - \sigma_3)^2 + (\sigma_1 - \sigma_3)^2\right] \tag{11}$$

By $F = \alpha I_1 + \sqrt{J_2}$, we get:

$$\frac{1}{D} = \exp\left[\left(\frac{F}{F_0(1-D)}\right)^m\right] \tag{12}$$

When $D = 0$, the soil is not damaged; $F = 0$ is a prerequisite that the equation mentioned above is valid.

During triaxial compression, the clay was subjected to stresses in three dimensions. There is $\alpha > 0$ in the Druck–Prager damage criterion; a necessary condition is followed as:

$$\sigma_1 = \sigma_2 = \sigma_3 = 0 \tag{13}$$

By analyzing the stress–strain curve, it can be seen that the stress–strain can be regarded as a linear relationship at the beginning of axial stress loading [35]. As the stress increases, the slope starts to get smaller and smaller at a certain point, which is nonlinear. It is possible to identify the specific point in the stress–strain curve for clay at which damage occurs. The stress value that corresponds to this point can be considered as the damage stress. So, Equation (13) derived above is unreasonable. If the stress is below the stress threshold corresponding to a certain point, the stress–strain curve shows a linear relationship, and it can be inferred that no damage happens in the clay at this point. Once the stress point surpasses the damage threshold value, damage emerges within the clay body, and it intensifies throughout the loading process.

Combining Equation (1) with Equation (9), the stress–strain relationship of the soil under triaxial stress can be expressed as:

$$\sigma = E\varepsilon_1 \exp\left[-\left(\frac{F^*}{F_0}\right)^m\right] + 2\mu\sigma_3 \tag{14}$$

$$\sigma = E\varepsilon_1 \exp\left[-\left(\frac{\alpha I_1 + \sqrt{J_2}}{F_0}\right)^m\right] + 2\mu\sigma_3 \tag{15}$$

where $E$ is the elastic modulus of soil, $\varepsilon_1$ is the axial strain of soil, $\mu$ is the Poisson ratio of soil, and $\sigma_3$ is the surrounding pressure of soil.

When considering the damage caused by freeze–thaw action, the freeze–thaw damage variable can be depicted by using a macroscopic scale. The damage variable incurred after a certain number of freeze–thaw experiences can be expressed as:

$$D_n = 1 - \frac{E_n}{E_0} \tag{16}$$

where $E_0$ is the elastic modulus of the soil without freeze–thaw cycles, and $E_n$ is the elastic modulus of the soil with freeze–thaw cycles of n times.

Zhang et al. [36] proposed that the concept of the benchmark damage state is that the soil is initially damaged under the external loading. As the damage intensifies, the

constitutive relationship of the soil damage under the combined effects of freeze–thaw action and loading can be defined as:

$$\sigma = E\varepsilon(1 - D_m) \tag{17}$$

$$D_m = D + D_n + DD_n \tag{18}$$

where $D_m$ is the damage variable under the combined loading and freeze–thaw action; $D$ is the damage variable under loading; $D_n$ is the damage variable under freeze–thaw actions alone.

Substituting Equations (16) and (18) into Equation (8), we get:

$$D_m = 1 - \frac{E_n}{E_0} \exp\left[-\left(\frac{\alpha I_1 + \sqrt{J_2}}{F_0}\right)^m\right] \tag{19}$$

Uniting Equations (15), (17), and (19), we get the damage constitutive equation under freeze–thaw actions and loading as:

$$\sigma = E_n\varepsilon \exp\left[-\left(\frac{\alpha I_1 + \sqrt{J_2}}{F_0}\right)^m\right] + 2\mu\sigma_3 \tag{20}$$

*4.3. Determination of the Parameters of the Damage Constitutive Equation*

According to the identity $\sqrt{J_2} = sI_1 + t$, we can get the parameters of this identity in Table 3.

**Table 3.** The linear relationship coefficients of $\sqrt{J_2} \sim I_1$.

| The Content of Moisture $\omega$ (%) | $s$ | $t$ | $R^2$ |
|---|---|---|---|
| 14 | 0.2254 | 132.145 | 0.9845 |
| 17 | 0.1647 | 112.014 | 0.9756 |
| 21 | 0.1225 | 97.014 | 0.9874 |
| 24 | 0.1014 | 80.257 | 0.9565 |

According to Table 3 above, the relationship of $s$ and $t$ and the content of moisture $\omega$ are obtained as follows:

$$s = 0.6745 \exp(-0.0474\omega) \tag{21}$$

$$t = 589.47\omega^{-0.48} \tag{22}$$

The damage conditions of deep expansive clay can be expressed as follows:

$$G = \sqrt{J_2} - sI_1 - t \tag{23}$$

Combining Equations (21), (22), and (23), we get the damage conditions of deep expansive clay; that is:

$$G = \sqrt{J_2} - 0.6745 \exp(-0.0474\omega)I_1 - 589.47\omega^{-0.48} \tag{24}$$

According to the damage threshold value relationship, the damage evolution equation of the damage variable can be obtained as:

$$D = \begin{cases} 0 & G < 0 \\ 1 - \exp\left[-\left(\frac{\alpha I_1^* + \sqrt{J_2^*} - F_G^*}{F_0}\right)^m\right] & G \geq 0 \end{cases} \tag{25}$$

where $F_G^*$ is the strength value of deep expansive clay of the damage valve.

By combining Equations (1), (5), (20), (24), and (25), the damage constitutive equation of deep expansive clay, considering the damage threshold value under freeze–thaw cycle conditions, can be expressed as:

$$\varepsilon = \begin{cases} \dfrac{\frac{\sigma-2\mu\sigma_3}{E_n}}{} & G < 0 \\ \dfrac{\sigma-2\mu\sigma_3}{E_n}\exp\left[\left(\dfrac{(\alpha I_1+\sqrt{J_2}-(\alpha+0.6745e^{-0.0474\omega})I_{1G}-589.47\omega^{-0.48}P_a)E\varepsilon_1}{F_0(\sigma-2\mu\sigma_3)}\right)^m\right] & G \geq 0 \end{cases} \tag{26}$$

where $E_n$ and $\mu$ are the elastic modulus and the Poisson ratio, respectively, under freeze–thaw cycles of n times.

The above expression can be simplified as follows:

$$\text{In}a + m\text{In}b = \text{In}\left[-\text{In}\left(\frac{\sigma-2\mu\sigma_3}{E_n\varepsilon}\right)\right] \tag{27}$$

where

$$a = \left(\frac{1}{F_0}\right)^m \tag{28}$$

$$b = \frac{\left[\alpha I_1 + \sqrt{J_2} - \left(\alpha + 0.6745e^{-0.0474\omega}\right)I_{1G} - 589.47\omega^{-0.48}P_a\right]E_n\varepsilon_1}{(\sigma-2\mu\sigma_3)} \tag{29}$$

The above expression can be simplified further as follows:

$$Y = mX + c \tag{30}$$

where $Y = \text{In}\left[-\text{In}\left(\frac{\sigma-2\mu\sigma_3}{E_n\varepsilon}\right)\right]$; $X = \text{In}b$; $c = \text{In}a$.

The values of $m$ and $c$ can be obtained by linearly fitting the data measured in the experiment to the above formula. According to $F_0 = \exp(-\frac{c}{m})$, $F_0$ can be obtained as follows in the table. The Poisson ratio of the deep expansive clay is taken to be 0.45, and then, based on the Druck–Prager expression of failure criterion, $\alpha I_1 + \sqrt{J_2} = k$, the value of $\alpha$ and k can be obtained by linearly fitting.

After the mechanical parameters of the damage model under the combined action of freeze–thaw cycles and loading were determined, they were listed in Table 4; the stress values under three confining pressures (100 kPa, 200 kPa, and 300 kPa) can be determined by using the final damage constitutive model. The measured data with a water content of 17% and six freeze–thaw cycles were compared with the theoretical data. As shown in Figure 7, when the confining pressure was 100 kPa, 200 kPa, and 300 kPa, the actual strength values were 118.4 kPa, 152.3 kPa, and 184.1 kPa, respectively, and the theoretical strength values are 120 kPa, 150 kPa, and 186 kPa. The fitting degree of the strength value was as high as 99%. The theoretical curve was basically consistent with the measured curve. The model can reasonably predict the damage evolution of soil under the combined action of loading and freeze–thaw cycles.

**Table 4.** The mechanical parameters of the damage model under the combined action of freeze–thaw cycles and loading.

| The Number of Freeze–Thaw Cycles | $\sigma_3$ = 100 kPa | | | $\sigma_3$ = 200 kPa | | | $\sigma_3$ = 300 kPa | | |
|---|---|---|---|---|---|---|---|---|---|
| | $E_n$ (MPa) | $m$ | $F_0$ | $E_n$ (MPa) | $m$ | $F_0$ | $E_n$ (MPa) | $m$ | $F_0$ |
| 0 | 647 | 0.384 | 0.457 | 674 | 0.341 | 0.445 | 660 | 0.316 | 0.471 |
| 3 | 584 | 0.324 | 0.387 | 485 | 0.387 | 0.381 | 543 | 0.384 | 0.249 |
| 6 | 502 | 0.318 | 0.754 | 534 | 0.345 | 0.241 | 526 | 0.365 | 0.426 |
| 9 | 424 | 0.336 | 0.714 | 496 | 0.319 | 0.674 | 507 | 0.314 | 0.874 |
| 12 | 388 | 0.329 | 0.646 | 403 | 0.327 | 0.429 | 469 | 0.352 | 0.773 |

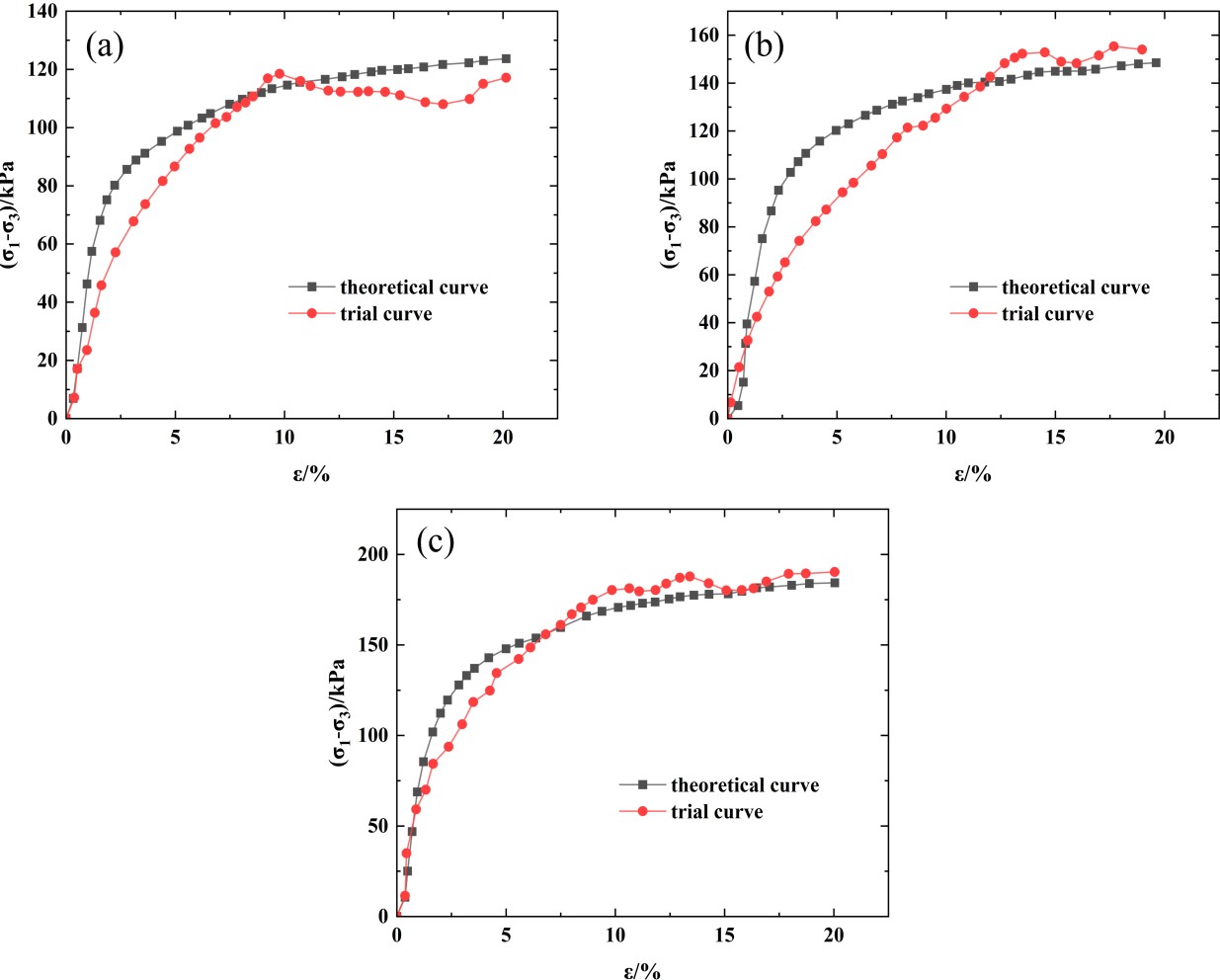

**Figure 7.** The comparison of the damage theoretical curve and test curve of deep expansive clay under different confining pressures. (**a**) 100 kPa, (**b**) 200 kPa, and (**c**) 300 kPa.

## 5. Discussion

In the early stage of this study, the conventional triaxial test and the triaxial test under freeze–thaw cycles were carried out on the deep expansive clay samples, and the mechanical properties of the soil were analyzed.

With the increase in the number of freeze–thaw cycles, the internal cracks of the soil will continue to expand with the expansion of water ice, which has a huge impact on the structure of the soil. How to properly reflect the damage law of soil under the combined action of loading and freeze–thaw cycles, the key is to select the appropriate damage variable. Many scholars had selected micro-benchmarks for damage analysis under loading [30,31]; however, there were some defects. The reason is that the initial soil damage has a great degree of randomness and random evolution. If the constitutive relationship of traditional methods is used, the randomness of soil in the damage process cannot be well reflected.

The reason is that the changes of microstructure and mechanical properties belong to micro and macro aspects, respectively. In order to study this problem fundamentally, it is necessary to combine macro research methods with micro research methods. In order to combine macro and micro research methods, macro and micro experiments must be carried out simultaneously. On this basis, this paper reasonably processes the test data by using probability and statistics methods, introduces appropriate damage variables, and establishes a damage constitutive theory combining the macro and micro.

Based on the research experience of previous scholars, this paper further studies the various mechanical properties of deep expansive clay and establishes the corresponding damage constitutive equation. However, due to time and other factors, the article still has shortcomings and needs further improvement:

1.  This paper mainly studies some properties of deep expansive clay, but it does not take into account the great differences in particle size, mineral composition, and structure of expansive clay at different depths. We can further explore the influence of these factors on the mechanical properties of soil.
2.  Whether the artificial soil can represent the deep environment and whether it is different from the deep in situ soil samples needs further discussion.
3.  Due to the limited amount of undisturbed soil and the test period and test equipment and other factors, the test sample used in the article is slightly insufficient; it can be supplemented by some relevant tests to make the conclusions of the article more reliable.

## 6. Conclusions

In this study, the conventional triaxial test and the triaxial test under freeze–thaw cycles were carried out on the deep expansive clay samples, and the mechanical properties of the soil were analyzed. Based on the test, the damage classification and research methods were discussed, and the damage constitutive equation of the combined action of freeze–thaw cycles and loading was determined by establishing appropriate damage variables.

The major conclusions of this study are as follows:

1.  In conventional triaxial test, the compressive strength of deep expansive clay gradually decreased with an increase in the water content. At this time, the stress–strain curve of clay under high confining pressure tended toward strain hardening, while low confining pressure shows strain softening.
2.  In the triaxial shear test under freeze–thaw cycles, the growth rate of stress with strain gradually decreased as the number of freeze–thaw cycles increased. Moreover, the ultimate peak stress also decreased as a result of the freeze–thaw cycles. Under different freeze–thaw cycles, the stress–strain curves of the triaxial tests all showed strain hardening; as the number of freeze–thaw cycles increased, the cohesion tended to decrease, while the internal friction angle tended to increase.
3.  Based on Lenaitre's strain equivalence hypothesis and the Druck–Prager damage criterion, the parameters of the damage constitutive equation are calculated and determined, and the calculated data are substituted into the final damage constitutive equation for verification. The fitting degree between the calculated strength value and the theoretical strength value is as high as 99%.
4.  The damage constitutive equation can reasonably predict the damage evolution of soil under the combined action of loading and freeze–thaw cycles. This study can serve as an available reference for well wall construction and disaster prediction in deep coal mining.

**Author Contributions:** Conceptualization, Z.Z. and S.C.; methodology, Z.Z. and S.C.; validation, Z.Z.; formal analysis, Z.Z.; investigation, S.C.; resources, B.L.; data curation, Z.Z. and S.C.; writing—original draft preparation, Z.Z. and S.C.; writing—review and editing, Z.Z.; visualization, Z.Z.; supervision, Z.Z.; project administration, B.L.; funding acquisition, B.L. All authors have read and agreed to the published version of the manuscript.

**Funding:** This research was funded by the University Doctoral Program Special Project, grant number 200803610004. The study was supported by the Natural Science Foundation of Anhui Province, grant number 2008085ME143.

**Data Availability Statement:** The data used to support the findings of this study are available from the corresponding author upon request.

**Conflicts of Interest:** The authors declare no conflict of interest.

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
