# Peer review of "Study on Mechanical Properties of Deep Expansive Soil and Coupling Damage Model of Freeze–Thaw Action and Loading"

_applsci, doi:10.3390/app131911099_

Round 1

Reviewer 1 Report

The paper presents a study related to soil characteristics changes when a process of freeze-thaw is applied. The used methods are correct. The paper clearly presents all the stages of the study conducted. There are no major changes needed. Only a careful editing is required as there are some minor language mistakes and redundant wording.

English quality is generally good. There are minor corrections needed.

Reviewer 2 Report

I recommend the publication of the current manuscript in your valuable journal (Applied Sciences) after major revisions:

1) Please highlight the novelty of the manuscript,

2) More details about the experimental conditions for all figures should be given and described,

3) Add the significations and the units to all abbreviations cited in the text to the nomenclature section (F-T, TSZ…..),

4) Are your all results, presented in all tables of the manuscript, the average of different runs? If yes what about the standard deviation in order to confirm the obtained results,

5) Figure 5 should be repeated and presented with high quality and resolution (It is very difficult to read number and legend in this figure),

6) It is better to present the units in all tables between brackets in the place of this / (for example in table 3 The content of moisture should be ω (%) in the place of ω/%

7) The sentences in the following line should be checked, conformed and compared to the literature review:

- Line 148…….due to internal damage,

- Line 189……due to the repeated freeze-thaw action, the small cracks in the soil,

- Line 207…..Due to the dispersion of soil particles, the soil is also,  

8) All the chemicals and equipment used in the experiments must be listed and where purchased. For the chemicals, indicate the degree of purity,

9) Compare and thoroughly discuss your results with the literature,

10) Recommendations and perspectives are missing,

11) The English language of the manuscript should be re-approved by a native speaker.

The English language of the manuscript should be re-approved by a native speaker

Reviewer 3 Report

Based on consolidated undrained triaxial tests on deep expanded clay, this article proposed the damage constitutive equation under the combined action of confining pressure and freeze-thaw (F-T) cycles, and obtained many reliable conclusions. It is also proposed that by combining the expression method of damage variables under separate load with the expression method under separate freeze-thaw cycle, this paper introduces the damage evolution equation and obtains the damage constitutive equation under the combined action of confining pressure and freeze-thaw cycle. The article is rich in content and innovative. It is recommended to be accepted after modification. Some problems in the article are recommended to be modified and explained, as listed below:

1. In line 113, pay attention to the writing of the table name, it should be "Table 1." Also make sure that the test described in lines 116 to 119 is equivalent to the test described in line 125. If it is the same experiment, but the loading rates described twice in the article are not consistent.

2. In line 172, in the description of the Mohr stress circle envelope diagram, by observing Figure 5, the internal friction angle does not seem to have an increasing trend. The slope of the tangent polyline to the Mohr stress circle can be marked in the figure, or the polyline and x The angle between the axis and the horizontal line. to reflect the change pattern, which may be more intuitive. If there are other ways, please explain. In addition, Figure 5 is relatively blurry, so it is recommended to increase the pixels and enlarge it.

3. This article does not have a Chapter 4. Please check the title number carefully. In addition, in 3.3, do you need to expand the stress and strain analysis of other water contents and sort out the mechanical characteristics such as compressive strength of all water contents to demonstrate the proposed The applicability and reliability of the damage evolution equation. The above are some of the problems I found in the article. This article adds the freeze-thaw cycle to the original damage evolution equation, introduces the damage evolution equation, and obtains the damage constitutive equation under the combined action of confining pressure and freeze-thaw cycles. It has been verified and fits the actual effect, but it is recommended to supplement and expand the validation section. Finally, I think the article is very good and very meaningful.

Reviewer 4 Report

1.    What are the differences of mechanical properties between deep expansive soil and non-deep soil? What are the interesting results of this study?

2.    The purpose of the moisture study is not stated. How is the moisture content determined? Does spraying water on the surface of the soil for a period of time ensure that the water and the soil mix evenly?

3.    Please explain the principle or basis of the experimental particle size setting in Table 2.

4.    The implication of the TSZ-2 in the abstract is incomprehensible. The first occurrence should be explained in detail.

5.    Are the results of figures b and c in Figure 5 correct? Is there any difference in order?

6.    The innovation of this research should be clear.

7.    The introduction is further organized to highlight innovation.

8.    Conclusions need to summarize the main points and results.

9.    What does the constitutive model do by itself? Please clarify.

Round 2

Reviewer 2 Report

Accept in present form

Accept in present form

Author Response

Dear Reviewer

    Thank you for your recognition of our work, we will continue to work hard to make better writing. I wish you good health and happiness.

Reviewer 4 Report

The conclusions should be concise and highlight the main findings.
